# Simple Carbohydrate Derivatives Diminish the Formation of Biofilm of the Pathogenic Yeast *Candida albicans*

**DOI:** 10.3390/antibiotics9010010

**Published:** 2019-12-30

**Authors:** Olena P. Ishchuk, Olov Sterner, Ulf Ellervik, Sophie Manner

**Affiliations:** 1Department of Biology, Lund University, Sölvegatan 35, SE-223 62 Lund, Sweden; ishchuk@chalmers.se; 2Centre for Analysis and Synthesis, Centre for Chemistry and Chemical Engineering, Lund University, P.O. Box 124, SE-221 00 Lund, Sweden; Olov.Sterner@chem.lu.se (O.S.); ulf.ellervik@chem.lu.se (U.E.)

**Keywords:** fucoside, xyloside, *Candida albicans*, biofilm

## Abstract

The opportunistic human fungal pathogen *Candida albicans* relies on cell morphological transitions to develop biofilm and invade the host. In the current study, we developed new regulatory molecules, which inhibit the morphological transition of *C. albicans* from yeast-form cells to cells forming hyphae. These compounds, benzyl α-l-fucopyranoside and benzyl β-d-xylopyranoside, inhibit the hyphae formation and adhesion of *C. albicans* to a polystyrene surface, resulting in a reduced biofilm formation. The addition of cAMP to cells treated with α-l-fucopyranoside restored the yeast-hyphae switch and the biofilm level to that of the untreated control. In the β-d-xylopyranoside treated cells, the biofilm level was only partially restored by the addition of cAMP, and these cells remained mainly as yeast-form cells.

## 1. Introduction

A majority of human microbial infections involve the formation of biofilms. Biofilms are cell communities that adhere to solid surfaces. Apart from forming biofilms on host surfaces, pathogens often form biofilms on abiotic surfaces of host implants, which brings serious implications for the use of medical devices such as prostheses, joint replacements, catheters, and pacemakers. Compared to planktonic cells, cells in biofilms are more resistant to antibiotics and antifungal drugs [1,2,3,4,5,6,7], host immune responses [8,9], and can promote further pathogen dissemination, forming new biofilms, which are associated with chronic or life threatening infections [10,11,12,13]. Thus, it is important to develop new strategies for the prevention of biofilm formation, and to eliminate already formed biofilm. 

*Candida albicans* is one of the most prevalent human opportunistic fungal pathogens and is able to form biofilm. Although it is a common part of the human microbiome [14], this yeast can, under certain conditions, cause various diseases. Mucosal infections such as vulvovaginal candidiasis occur in up to 75% of all women [15]. Furthermore, immune-compromised individuals may develop candidiasis of the oral cavity as well as systemic blood stream infections with high mortality [16]. The biofilms developed by *C. albicans* are highly heterogenous and made up of a dense matrix, composed of morphologically different cell types (i.e., blastospores (yeast-form cells), hyphae, and pseudo-hyphae cells surrounded by extracellular polymeric substances (EPS)) [1,17]. The basal thin biofilm layer is composed of yeast-form cells followed by a thicker and more open hyphae layer [1,18,19,20]. In addition to being a scaffold component, the hyphae development is an important virulence factor and facilitates the pathogen invasion through epithelial tissue of the host [21]. The cell wall of *C. albicans* hyphae and yeast-form cells differ in its composition, which affects their adhesion and recognition by the host’s immune system. For example, the hyphal glucan consists of a unique cyclic (1-3)-linked polymer backbone with long (1-6)-linked side chains [22]. Hyphae also have less mannan, compared to yeast-form cells [23]. The morphological transition from yeast-form cells to hyphae (i.e., switching) is a complex, tightly regulated process during the biofilm maturation, which is regulated by environmental conditions and quorum sensing molecules such as farnesol and tyrosol [21]. Multiple signaling pathways such as cyclic adenosine monophosphate/protein kinase A (cAMP-PKA), high-osmolarity glycerol (HOG), mitogen-activated protein kinases (MAPK), and several transcriptional factors, Efg1, Bcr1, Ndt80, Cph1, Cph2, Tec1, Nrg1, Rfg1, and Tup1 are known to regulate the morphological transition and hyphal adhesion [1,21,24,25,26]. Upon stimuli favoring hyphae, the yeast cells produce a germ tube that extends by polarized growth by the activity of membrane-bound vesicles supplying the components of the plasma membrane and cell wall to the growing hyphae [21].

Different carbohydrates have been explored for their possibilities to disturb or prevent adhesion, biofilm formation, and the extension of an already existing biofilm of *C. albicans* [27,28,29,30,31,32,33,34]. In addition, some natural products of plant origin were recently shown to reduce the adhesion of *C. albicans* and its biofilm formation by inhibiting the yeast-to-hyphae transition [35,36].

In this study, we synthesized and tested the effect of simple monosaccharides and glycosides ([Fig antibiotics-09-00010-ch001]) on *C. albicans* biofilm development on polystyrene surfaces.

## 2. Results

### 2.1. Carbohydrate Derivatives

The unprotected carbohydrates (**1a**, **2a**, **3a**, and **4a**) and the methyl pyranosides (**1c**, **2c**, and **3c**) were commercially available. The synthesis of the benzylated compounds **1b** [37], **2b** [38], and **3b** [39] was performed as described previously. **4a** was benzylated at the anomeric position using benzyl alcohol and acetyl chloride [40] and deprotected to give compound **4b** in a 32% yield over two steps (Scheme 1). Compound **4b** was synthesized previously via dihydroxylation of the corresponding deoxy sugar [41].

### 2.2. Biofilm Formation

To evaluate the effects of carbohydrate derivatives, the formation of biofilm on polystyrene surfaces was measured using liquid cultures with the *Candida albicans* SC5314 strain and the addition of 10 mg/mL of each compound. The formation of biomass and biofilm was measured after 24 h of incubation using the crystal violet method [42]. The data are shown in Figure 1.

During the biofilm experiments, the *C. albicans* biomass substantially increased only on mannose (**1a**) (2.3 times, Analysis of Varience (ANOVA) *p* = 0.0079), but not on the other carbohydrates. None of the tested compounds inhibited biomass except for **1c**, which reduced the biomass formation by ~40%, ANOVA *p* = 0.0225 (Figure 1). The unprotected monosaccharides **1a**–**4a** and the methyl glycosides **1c**–**3c** did not show any significant effect on the formation of biofilm. In contrast, the benzyl-glycosides reduced the formation of biofilm. After 24 h of incubation, compounds **2b**, **3b**, and **4b** reduced the biofilm 2.1, 7.1, and 4.3 times, respectively (ANOVA *p* = 0.0491; 0.0029 and 0.0039), compared to the untreated control (Figure 1).

Compounds **3b** and **4b** were selected for further analysis due to their highest inhibition effect on biofilm. The viability of biofilms of *C. albicans* after treatment with **3b** and **4b** at different concentrations was evaluated by staining the adherent cells with XTT (2, 3-bis (2-methoxy-4-nitro-5-sulfophenyl)-5-[(phenylamino)carbonyl]-2H-tetrazolium hydroxide), a colorimetric assay for the quantification of cellular viability and cytotoxicity [43]. Compounds **3b** and **4b** each displayed a dose-dependent inhibitory effect on biofilm viability (Figure 2a). The minimum biofilm inhibitory concentration (MBIC) was estimated to be 5 mg/mL and resulted in 3.7-times reduced biofilm (Figure 2a, ANOVA *p* = 0.03 for both **3b** and **4b**).

The addition of cAMP (regulatory molecule of major signaling pathway of biofilm development, the cAMP-PKA pathway) to the media with carbohydrates affected the biofilm development. It rescued the inhibition by the **4b** compound and restored the biofilm level to that of the untreated control (Figure 2b). On the other hand, the inhibitory effect of **3b** was only partially restored by cAMP treatment (Figure 2b).

### 2.3. Yeast-Hyphae Transition

The formation of hyphae by *C. albicans* is important for biofilm formation since hyphae are more adherent than yeast cells. After the formation of hyphae, the adhesins, which are expressed mostly on hyphae, play an important role in further adhesion [9]. To monitor the *C. albicans* cell morphology upon carbohydrates treatment, we also studied the biofilm development using microscopy and microfluidics.

After the yeast cells were inoculated into the biofilm medium, hyphae started to form within the first hour of incubation (YNB supplemented with 0.45% ammonium sulphate, 100 mM l-proline and 0.2% glucose, pH 7.0, see Figure 3a, Appendix A, time-lapse videos of biofilm formation over time). The addition of benzyl β-d-xylopyranoside (**3b**) or benzyl α-l-fucopyranoside (**4b**) resulted in a substantial decrease of hyphae formation, and the majority of the cells remained in yeast-form during the incubation period (Figure 3b,c, Appendix A, time-lapse videos of biofilm formation over time). High-resolution pictures after 10 h and 18.5 h of incubation can also be found in the Appendix A.

The branching of hyphae and cell size could be affected by vacuole inheritance [44,45]. However, the benzylated compounds tested did not exhibit any effect on vacuole morphology. Furthermore, the addition of cAMP to the media released the hyphae formation inhibitory effect caused by benzyl α-l-fucopyranoside (**4b**) (Appendix A) and the switching of yeast-form cells to hyphae occurred effectively. In contrast, the cAMP did not rescue the benzyl β-d-xylopyranoside (**3b**) inhibition effect on hyphae formation, and most cells remained in yeast-form after 24 h of the treatment (Appendix A).

## 3. Discussion

In the current study, we developed new carbohydrates derivatives that diminished the biofilm formation of an opportunistic pathogen *Candida albicans.* Carbohydrates are abundant in nature and are a part of the building blocks of living cells. Xylose and fucose are found in the fungal extracellular biofilm matrix [46]. Xylose comprises around 12% of total carbohydrates of *C. albicans* biofilm matrix [46]. Furthermore, l-fucose is a component of host glucan surface structures, found in mammalian cells. The *C. albicans* Als1 protein specifically recognizes the fucose-containing sugars of glucan of the host [47]. Thus, the addition of these carbohydrates can possibly perturb the pathogen interaction with the host or competitively bind the pathogen adhesins and prevent adhesion to abiotic surfaces. Therefore, these carbohydrates have been investigated for their ability to inhibit adhesion and biofilm formation.

In the early 1980s, the effect of both fucose and xylose was studied on *C. albicans* adhesion to human cells. Sobel et al. showed that both d- and l-fucose inhibited *C. albicans* adhesion to vaginal epithelial cells (by ~30%) [27]. Xylose inhibited the adhesion to buccal cells by 15% [28]. In our study, the addition of l-fucose and d-xylose to the biofilm media (containing the hyphae inducer l-proline) slightly decreased the biofilm formation on polystyrene. On the other hand, the benzyl-derivatives of both fucose and xylose had a significantly higher inhibitory effect on the biofilm formation of *C. albicans*. We found that in addition to reducing the adhesion to polystyrene, these compounds affected the *C. albicans* morphological transition of yeast-form cells to hyphae, which is the key property for mature biofilm [1]. Upon the addition of either benzyl α-l-fucopyranoside (**4b**) or benzyl β-d-xylopyranoside (**3b**), the cells continued to grow in yeast-like form and produced only few hyphae. On the other hand, methyl glycosides did not affect the morphological transition, indicating that the inhibitory effect is linked to the benzyl group. The morphological transition of yeast to the hyphae form is regulated by the cAMP-PKA pathway in *C. albicans* [1,25]. Interestingly, aromatic compounds have been reported to influence the adhesion of both *Saccharomyces cerevisiae* and *C. albicans* by inducing the expression of adhesin genes [48,49]. For *S. cerevisiae*, tryptophol and phenyl ethanol were reported to be a part of quorum sensing by activating the *FLO11* gene expression via the cAMP-PKA pathway [48,50]. *C. albicans* was found to respond to the aromatic alcohol tyrosol [51], while tryptophol and phenylethanol did not induce phenotypic changes in *C. albicans* [50]. Farnesol is a negative regulator of quorum sensing through the Ras-cAMP-PKA pathway in both *C. albicans* and *S. cerevisiae*, [49,52] as it was found to inhibit *C. albicans* adhesion [53] and *S. cerevisiae* cell growth [54].

The addition of cAMP (a key regulatory molecule in yeast-to-hyphae transition of Ras-cAMP-PKA pathway) to the biofilm medium containing benzyl-glycosides reversed the hyphae formation inhibitory effect only of benzyl α-l-fucopyranoside (**4b**), but not of benzyl β-d-xylopyranoside (**3b**). Apart from restoring the morphological transition, the cAMP restored the adhesion level of the α-l-fucopyranoside treated sample to 50% of the control. This suggests that the effects of benzyl α-l-fucopyranoside (**4b**) and benzyl β-d-xylopyranoside (**3b**) are mediated by different mechanisms. Compound **4b** likely operates through the Ras-cAMP-PKA pathway by inhibiting the pathway by downregulating the level of cAMP. Compound **3b** could affect the pathway gene expression downstream of cAMP. Future studies such as *C. albicans* transcriptome analysis upon the exposure to the compounds tested are needed to find the gene targets.

## 4. Materials and Methods

### 4.1. General Methods, Strains, and Growth Conditions

Thin-layer chromatography (TLC) was performed on precoated TLC-glass plates with silica gel 60 F_254_ 0.25 mm (Merck). Spots were visualized with UV light or by charring with an ethanolic anisaldehyde solution. Preparative chromatography was performed on a Biotage Isolera One flash purification system using Biotage SNAP KP-Sil silica cartridges. Nuclear magnetic resonance (NMR) spectra were recorded at ambient temperatures on a Bruker Avance II at 400 MHz (^1^H), and 100 MHz (^13^C) and assigned using 2D methods (COSY, HMQC). Chemical shifts are reported in ppm, with reference to residual solvent peaks (δH CHCl_3_ = 7.26 ppm and δC CDCl_3_ = 77.0 ppm). Coupling constant values are given in Hz. ^13^C-NMR spectra are proton decoupled. High-resolution mass spectra (HRMS) were recorded on Waters QTOF XEVO-G2 (ESI+).

*Candida albicans* SC5314 strain [55] was grown at 37 °C in complete medium YPD (0.5% yeast extract [Sigma-Aldrich, Y1625], 1% peptone [Sigma-Aldrich, 70175], 2% glucose [Sigma-Aldrich, G7021]) or minimal medium consisting of YNB (yeast nitrogen base without amino acids and ammonium sulfate, FORMEDIUM^TM^, CYN0505) supplemented with 0.45% ammonium sulfate (Sigma-Aldrich, A4915), 0.2% glucose (Sigma-Aldrich, G7021), and 100 mM l-proline (Sigma-Aldrich, P0380) of pH 7.0. If needed, 2% agar (Sigma-Aldrich, A5306) was used to solidify the media.

The liquid minimal medium (YNB [yeast nitrogen base without amino acids and ammonium sulfate, FORMEDIUM^TM^, CYN0505] supplemented with 0.45% ammonium sulfate [Sigma-Aldrich, A4915], 0.2% glucose [Sigma-Aldrich, G7021], and 100 mM l-proline [Sigma-Aldrich, P0380]) of pH 7.0 was used for the biofilm assay (biofilm medium).

Different carbohydrates ([Fig antibiotics-09-00010-ch001]) were added to the biofilm medium at the final concentration 0.1–15 mg/mL. The cell permeable cAMP (N^6^, 2′-*O*-Dibutyryladenosine 3′,5′-cyclic monophosphate sodium salt, Sigma-Aldrich, D0260) was added to the biofilm medium at the final concentration of 10 mM to study if the sugar inhibition effect on the biofilm could be reversed.

### 4.2. Biofilm Assay

Prior to the biofilm assay, yeast cultures were grown in liquid YPD medium for 24 h until the stationary phase (OD_600_ 11–17), cells were then pelleted by centrifugation (1699 *g*), washed with sterile water, and cells were further inoculated into a test biofilm medium at a final concentration of 0.2 OD_600_/mL and incubated in 96-well flat-bottom polystyrene microtiter plates (Sigma-Aldrich, Corning^®^ Costar^®^ culture plates, CLS3596-50EA) for 24 or 48 h at 37 °C. At the defined time points, the biofilm developed on polystyrene was measured either by crystal violet staining as described [42,56] or by improved XTT method, where the 200 mM glucose and XTT (2, 3-bis (2-methoxy-4-nitro-5-sulfophenyl)-5-[(phenylamino)carbonyl]-2H-tetrazolium hydroxide, X4626, Sigma-Aldrich) were added to the reaction mixture [43]. The crystal violet (HT901-8FOZ, Sigma Aldrich) was added to the media at the final concentration of 0.05%. After 24 and/or 48 h of cell staining, plate wells were washed four times with 200 μL of water to remove planktonic cells, and the biofilms were then dried and dissolved in 200 μL of 96% ethanol. In the XTT assay, the planktonic cells were removed by washing twice with PBS buffer. Total biomass and crystal violet biofilm staining measurements were performed at OD_560_, and the formazan formed during the XTT biofilm assay was measured at OD_485_ with FLUOstar OPTIMA plate reader, BMG LABTECH.

### 4.3. Microfluidics and Microscopy

Microfluidic plates (CellASIC^®^ ONIX, Merck Millipore, Y04D-02-5PK) were used with the ONIX Microfluidic Perfusion System and inoculated with yeast at 8 psi for 5 s, according to the manufacturer’s recommendations, and flowed at 1.5 psi with media tested (with/without carbohydrates, 15 mg/mL). The yeast growth and biofilm development were monitored over time on a fully motorized and automated inverted widefield microscope Observer Z1 (Carl Zeiss) equipped with a sCMOS camera. The phase-contrast images were taken over the time specified.

### 4.4. Synthesis of Benzyl (2,3,4-tri-O-acetyl) α-l-fucopyranoside (5)

Acetyl chloride (0.43 mL, 6.1 mmol) was added to a stirred solution of benzyl alcohol (8.83 mL, 85.0 mmol) and l-fucose **4a** (1.0 g, 6.1 mmol). After 22 h at 50 °C, the reaction mixture was allowed to reach room temperature (r.t.) before the removal of benzyl alcohol by vacuum distillation (0.1 torr, 65 °C). The dark brown crude residue was dissolved in pyridine (9.0 mL, 0.11 mol) and acetic anhydride (9.0 mL, 0.16 mol) and stirred at r.t. After 15 h, toluene was added, and the solvent was removed under reduced pressure. Purification by column chromatography (Biotage, KP-Sil 50 g, heptane:EtOAc 95:5→88:12) gave benzyl (2,3,4-*tri*-*O*-acetyl) α-l-fucopyranoside (**5**, 986 mg, 56%, clear oil) and benzyl (2,3,4-*tri*-*O*-acetyl) β-l-fucopyranoside (**6**, 247 mg, 14%, white amorphous solid). **5:**
^1^H-NMR (CDCl_3_) δ 7.37–7.28 (m, 5H) 5.42–5.38 (m, 1H) 5.30–5.29 (m, 1H) 5.15–5.12 (m, 2H) 4.71 (d_AB_, 1H, *J*_AB_ 12.4 Hz) 4.55 (d_AB_, 1H, *J*_AB_ 12.0 Hz) 4.17 (q, 1H, *J* 6.4 Hz) 2.16 (s, 3H) 2.03 (s, 3H) 1.98 (s, 3H) 1.11 (d, 3H, *J* 6.4 Hz); ^13^C-NMR (CDCl_3_) δ 170.7, 170.5, 170.2, 137.3, 128.6, 128.1, 127.9, 95.6, 71.3, 70.0, 68.2, 68.2, 64.7, 20.9, 20.8, 20.8, 15.9; HRMS (ESI/QTOF) *m*/*z* [M + Na] calcd. for C_19_H_24_O_8_Na 403.1363, found 403.1368. **6:**
^1^H-NMR (CDCl_3_) δ 7.37–7.28 (m, 5H) 5.28–5.22 (m, 2H) 4.98 (dd, 1H, *J* 3.6, 10.4 Hz) 4.92 (d_AB_, 1H, *J*_AB_ 12.4 Hz) 4.62 (d_AB_, 1H, *J*_AB_ 12.4 Hz) 2.18 (s, 3H) 2.00 (s, 3H) 1.98 (s, 3H) 1.25 (d, 3H, *J* 6.4 Hz); ^13^C-NMR (CDCl_3_) δ 170.9, 170.4, 169.7, 137.2, 128.5, 128.0, 127.8, 99.8, 71.5, 70.7, 70.4, 69.3, 69.1, 20.9, 20.9, 20.8, 16.2; HRMS (ESI/QTOF) *m*/*z* [M + Na] calcd for C_19_H_24_O_8_Na 403.1363, found 403.1369.

### 4.5. Synthesis of Benzyl α-l-fucopyranoside (4b)

**5** (0.18 g, 0.63 mmol) was dissolved in MeOH (3 mL) and 1 M NaOMe (0.15 mL, 0.15 mmol) was added. After 2 h, glacial acetic acid was added until neutral pH. The reaction mixture was co-evaporated with toluene several times. Purification by column chromatography (SiO_2_, CH_2_Cl_2_:MeOH 95:5) gave **4b** (90 mg, 57%) as a white amorphous solid. Data analysis was according to the published data [41].

### 4.6. Statistical Analysis

The software package Minitab^®^ 18.1 was used to analyze the obtained data.

## 5. Conclusions

We showed that benzyl β-d-xylopyranoside (**3b**) and benzyl α-l-fucopyranoside (**4b**) were both effective against biofilm and inhibited the morphological transition of *C. albicans* from yeast-form cells to hyphae, diminishing the biofilm. The MBIC for both β-d-xylopyranoside and α-l-fucopyranoside was estimated as 5 mg/mL. Neither the unprotected monosaccharides, nor the corresponding methyl-glycosides showed any significant effects on biofilm inhibition.

The exposure to cAMP of α-l-fucopyranoside treated cells restored yeast-hyphae switching and the biofilm level to that of the untreated control. While the biofilm level was partially restored by the addition of the cAMP, the benzyl β-d-xylopyranoside treated cells remained mainly in yeast-like form. We propose that the effects shown by these compounds are mediated by different mechanisms.

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
