# Peer review of "Simple Carbohydrate Derivatives Diminish the Formation of Biofilm of the Pathogenic Yeast Candida albicans"

_antibiotics, 2019, doi:10.3390/antibiotics9010010_

Round 1
Reviewer 1 Report
Biofilm formation is part of the ability of Candida albicans, a normal component of the human microbial community, to cause diseases. Biofilm formation has been tightly linked to the development of resistance to antifungal drugs therefore an intense research is developed in the search for compounds that can solve this problem.
This article presents the effect of some simple monosaccharides and glycosides on C. albicans biofilm development. Also, for the most active compounds, the effect on yeast-hyphae transition was studied. The results are encouraging and this work can be continued especially considering that the structure of the compounds is simple and has a facile synthesis route.
For the future work it will be useful for the authors to consider also the determination of some quantitative parameters as MBEC (minimum biofilm eradication concentration).
The references must contain some more recent articles, especially in the introduction part
.
Author Response
Comment: Biofilm formation is part of the ability of Candida albicans, a normal component of the human microbial community, to cause diseases. Biofilm formation has been tightly linked to the development of resistance to antifungal drugs therefore an intense research is developed in the search for compounds that can solve this problem.
This article presents the effect of some simple monosaccharides and glycosides on C. albicans biofilm development. Also, for the most active compounds, the effect on yeast-hyphae transition was studied. The results are encouraging and this work can be continued especially considering that the structure of the compounds is simple and has a facile synthesis route.
For the future work it will be useful for the authors to consider also the determination of some quantitative parameters as MBEC (minimum biofilm eradication concentration).
Response: According to the reviewer suggestion we estimated the minimum biofilm inhibitory concentration (MBIC) for both compounds and it is 5 mg/ml.
Comment: The references must contain some more recent articles, especially in the introduction part
Response: The introduction was revised and also more references added. There are only few publications regarding carbohydrates for biofilm treatment found.
Reviewer 2 Report
The manuscript describes the use of novel carbohydrate compounds (some of which that are commercially available), in order to restrict biofilm formation and yeast-hypha transition (which is subsequently rescued with addition of cAMP).
The manuscript is of general interest, but has some points that I feel need addressing.
The introduction is largely fine, but if the authors are able to within any word limits, an elaboration on processes related to morphological transition would be of interest to the readers.
Methods
I do not have sufficient expertise to verify the synthetic chemistry of the compounds, and it is complicated to follow in a general sense. If this can be re-worded for a more lay audience, this would be of benefit.
Culture medium - the 'biofilm medium' (L177) shows different conc of ammonium sulphate to L191. Please correct.
Supplier information is necessary for the YNB and YPD (if made in house, suppliers for the components).
No statistical analyses are given - this is very important.
Results
Fig 1. this shows biomass and 'biofilm'. It is not clear what 'biofilm' is related to. Is this metabolic activity? In which case it is not 'biofilm'. These are different analyses.
Further, although the results are normalised to 100% (control), there will be variation in the control too, so should be on the graph. What are the error bars?
The authors suggest there are no differences in biomass, nor 'biofilm', but there are quite large differences evident. Some near 50%. Statistical analyses will dictate whether the results are significant or not, not just by eye. Please perform statistical tests and update manuscript. (avoid use of 'significant' in text without stats data)
Fig 3 shows some clear differences between both biomass and qty of hyphae. This needs to be enumerated and presented as numeric data too. As only one image is shown, this could be the 'best case' image that shows what you want. The authors do not state whether one or five, or indeed ten+ images were taken, and a representative one selected (at random?). This will give more data to support your conclusions, and additional stats tests to confirm robustness.
What are the shapes in the middle of the images? These seem to get in the way of things.
Discussion is relatively short and although relates to the literature, does not suggest or propose any mechanisms by which the carbohydrates could be exerting an effect, despite showing a rescue state with cAMP. It is fine to blue-sky think for this section, but for others reading, future studies or suggestions of things would also be of interest.
Author Response
Comment: The manuscript describes the use of novel carbohydrate compounds (some of which that are commercially available), in order to restrict biofilm formation and yeast-hypha transition (which is subsequently rescued with addition of cAMP).
The manuscript is of general interest, but has some points that I feel need addressing.
The introduction is largely fine, but if the authors are able to within any word limits, an elaboration on processes related to morphological transition would be of interest to the readers.
Response: The introduction was revised. More information on morphological transition added.
Comment: Methods
I do not have sufficient expertise to verify the synthetic chemistry of the compounds, and it is complicated to follow in a general sense. If this can be re-worded for a more lay audience, this would be of benefit.
Response: We have used a standard description for synthesis of carbohydrates. Experimental details are provided so that an organic chemist can repeat the synthetic procedure.
Comment: Culture medium - the 'biofilm medium' (L177) shows different conc of ammonium sulphate to L191. Please correct.
Response: Corrected.
Comment: Supplier information is necessary for the YNB and YPD (if made in house, suppliers for the components).
Response: This information was added.
Comment: No statistical analyses are given - this is very important.
Response: Statistical analyses were added.
Comment: Results
Fig 1. this shows biomass and 'biofilm'. It is not clear what 'biofilm' is related to. Is this metabolic activity? In which case it is not 'biofilm'. These are different analyses.
Further, although the results are normalised to 100% (control), there will be variation in the control too, so should be on the graph. What are the error bars?
Response: Figure legend was rewritten. The biofilm (cell adherent to polystyrene) was assayed by crystal violet.
Comment: The authors suggest there are no differences in biomass, nor 'biofilm', but there are quite large differences evident. Some near 50%. Statistical analyses will dictate whether the results are significant or not, not just by eye. Please perform statistical tests and update manuscript. (avoid use of 'significant' in text without stats data)
Response: We have largely rewritten this section to include statistical analysis.
Comment: Fig 3 shows some clear differences between both biomass and qty of hyphae. This needs to be enumerated and presented as numeric data too. As only one image is shown, this could be the 'best case' image that shows what you want. The authors do not state whether one or five, or indeed ten+ images were taken, and a representative one selected (at random?). This will give more data to support your conclusions, and additional stats tests to confirm robustness.
Response: These data are used to support the quantitive ones obtained in Figure 2 (biofilm-cell adhesion measured to polystyrene surface). To help the reader, we added time-lapse vides of Fig 3 experiment as supporting information.
Comment: What are the shapes in the middle of the images? These seem to get in the way of things.
Response: These are the numbers of the chambers of microfluidics plate. This information was added to the figure legend.
Comment: Discussion is relatively short and although relates to the literature, does not suggest or propose any mechanisms by which the carbohydrates could be exerting an effect, despite showing a rescue state with cAMP. It is fine to blue-sky think for this section, but for others reading, future studies or suggestions of things would also be of interest.
Response: Discussion was revised. More information added about the regulation, for example, cAMP-PKA signaling pathway is the major regulator of yeast-hyphae transition.